# Parametric Characteristics and Bifurcation Analysis of Multi-Degree-of-Freedom Micro Gyroscope with Drive Stiffness Nonlinearity

**DOI:** 10.3390/mi10090578

**Published:** 2019-08-30

**Authors:** Mingjiang Han, Qichang Zhang, Shuying Hao, Weixiong Li

**Affiliations:** 1Tianjin Key Laboratory of Nonlinear Dynamics and Control, Department of Mechanics, School of Mechanical Engineering, Tianjin University, Tianjin 300072, China; 2Tianjin Key Laboratory of Advanced Electromechanical System Design and Intelligent Control, Tianjin University of Technology, Tianjin 300384, China; 3National Experimental Teaching Demonstration Center of Mechanical and Electrical Engineering, Tianjin University of Technology, Tianjin 300384, China

**Keywords:** micro gyroscope, multi-degree-of-freedom (MDOF), stiffness nonlinearity, internal resonance, bifurcation

## Abstract

The dynamic equations of a four-degree-of-freedom micro gyroscope system were developed considering the nonlinearity of driving stiffness, the primary resonance, and the 1:1 internal resonance. Then, the perturbation analysis was carried out using the method of multiple scales. The influence of stiffness nonlinearity and system parameters on micro-gyro dynamic characteristics, output sensitivity, detection bandwidth, and working stability were discussed based on the analytic and numerical solutions of the dynamic equations. Through the singularity theory, the influence of system parameters on bifurcation behavior was analyzed. The results show that the amplitude jump and multi-stable solutions caused by the nonlinear hardening characteristics of the high robust two-degree-of-freedom drive-mode occur outside the detection bandwidth. In addition, the influence on the bandwidth was weak and the sensitivity of the bandwidth area was slightly reduced. Moreover, saturation existed in the response amplitude of the second drive-mode in spite of the primary resonance being completely tuned or detuned. As a result, although the electrostatic force amplitude was out of the unstable region and even took a larger value, the micro gyroscope obtained a larger stable output. Besides, nonlinearity will lead to energy transfer between various modes of multi-degree-of-freedom micro gyroscopes. That means the response amplitudes could change greatly due to the variation of the external environment even the system is under a constant excitation frequency. Therefore, increasing the stiffness coefficient of the micro beam and the electrostatic force amplitude can maintain the robustness of the system to environmental changes and avoid the occurrence of bifurcation.

## 1. Introduction

The micro-electro-mechanical gyroscope is a kind of inertial device used to measure angular velocity and angular displacement, which is developed on the basis of micro-electro-mechanical system (MEMS) processing technology and measurement and control technology. It is widely used in many fields, such as aerospace, weapon guidance, automotive navigation, and robotics, because of its small size, low power consumption, reliability and stability, and batch production [1,2,3,4]. The drive and sense mode of the traditional micro-mechanical gyros mainly utilized the single-degree-of-freedom (1-DOF) structure, which increases the mechanical sensitivity by matching the natural frequencies of the drive and sense mode [5,6]. However, mode matching also leads to a significant reduction in bandwidth and robustness, making the micro gyroscope system susceptible to structural and environmental changes [7]. In order to solve the inherent design defects of 1-DOF micro gyroscopes, some multi-degree-of-freedom (MDOF) micro gyroscope structures [8,9,10,11] have been proposed by increasing the degree-of-freedom of the drive and sense mode. Among them, the 4-DOF micro gyroscope, which utilizes complete 2-DOF drive and sense mode, has attracted much attention because of its high robustness and wide bandwidth [12,13]. For example, Acar et al. [12] proposed a resonant-free 4-DOF micro gyroscope structure with large amplitude by means of dynamic amplification for the drive and sense mode. Consequently, the bandwidth and robustness of the drive and sense response were greatly enhanced. Furthermore, the vibration of the drive and sense direction was effectively decoupled due to the addition of the decoupling frame. Wang et al. [13] designed a novel 4-DOF micro gyroscope on the basis of Acar’s work, which achieved higher sensitivity while maintaining wide bandwidth. At the same time, the drive and sense output bandwidth were highly matched, which further improved the robustness of the micro gyroscope. This micro gyroscope could not only respond quickly to input signals but could also suppress noise signals effectively. Jia et al. [14] proposed a frequency tuning technology based on a quadrature modulation signal to eliminate the frequency mismatch of MEMS gyroscopes. The closed-loop detection method was adopted, which improved the scale factor nonlinearity and bandwidth under the premise of maintaining the same static performances compared with the open-loop detection by tuning. Ou et al. [15] proposed an oblique suspension beam, adopting a polygonal cross-section to enhance the sensitivity and robustness of the butterfly vibratory gyroscope (BFVG). The theoretical arithmetic results suggested that a polygonal cross-section beam are much more stable than a convex cross-section beam in most cases.

The electrostatic-driven MEMS gyroscope is a multi-field coupled nonlinear dynamical system, which contains many nonlinear factors, such as material nonlinearity, damping nonlinearity [16,17], electrostatic nonlinearity [18,19], and geometric nonlinearity caused by large deformation of micro-mechanical structural elements. Among them, nonlinear stiffness of elastic microbeam and electrostatic force nonlinearity are the most common issues in micro gyroscope systems. The existence of these nonlinear factors will lead to obvious frequency offset, multi-stable solution, and softening-hardening characteristics of stiffness and transition of soft and hard characteristics [20,21] of the micro gyroscopes, even oscillation instability such as bifurcation or chaos [22]. Yet, these effects have a significant influence on the sensitivity, bandwidth, and stability of micro gyroscopes. Therefore, the global dynamics and local bifurcation of the micro gyros high-dimensional nonlinear systems should be taken into account during the design. In other word, the complex dynamic behaviors and their influence on the sensitivity, bandwidth, and stability of the micro gyroscope should be revealed and used to guide, compensate, and control in order to reduce the influence of nonlinear factors. This is one of the key issues that must be solved to improve the dynamic performance of micro gyroscopes. Sharma et al. [23] have realized signal amplification, or the attenuation of undesired signal components, by adjusting the phase difference between driving force and parameter coupling. Kacem et al. [24] described a comprehensive nonlinear multi-physics model based on the Euler-Bernoulli beam equation that remains valid up to large displacements in the case of electrostatically actuated Mathieu resonators. Nitzan et al. [25] introduced the concept of self-induced parametric amplification, and perfect degeneracy of the primary and secondary vibration modes was achieved through electrostatic frequency tuning, which also enabled the phase and frequency of the parametric coupling to be varied. Their results showed that the resulting phase and frequency dependence of the amplification follow the theory of parametric resonance. Tsai et al. [26] established a nonlinear dynamic model of a class of three-axis vibrating wheel micro gyroscopes by considering nonlinear stiffness and nonlinear electrostatic force. The influence of system parameter variation on driving torque was theoretically analyzed, and the results indicated that the drive and sense modes have similar transition orbits in the transition of micro gyroscope motion modes in chaos analysis. Li et al. [27] studied the forced vibration of the gyro system under harmonic excitation and its dynamic behavior under the control of displacement and velocity feedback with time delay. The effects of system parameters and time delayed feedback gain on the system amplitude were discussed. Wang et al. [28] proposed a method using the stiffness nonlinearity of the 1-DOF micro gyroscope in the driving direction. The inherent hardening characteristics of the stiffness nonlinearity were used to match the resonant frequency to achieve a higher amplitude than the linear design. Using experiments and simulations, Xu et al. [29] verified that the amplitude-frequency curves of the gyroscope have a wide flat region and a higher amplitude than the linear design when the driving direction is nonlinear. A new driving method was proposed to excite the large amplitude nonlinear vibration in the downward scanning characteristic curve. Lajimi et al. [30] presented the nonlinear dynamical features of a gyroscopic system manifesting in a rotation rate sensor. A computational shooting method and Floquet multipliers were used to characterize the response. The study showed that larger bandwidth and higher sensitivity appeared when the system operating in the nonlinear regime. Wen et al. [31] investigated the design principle of the detection bandwidth in micro gyros when nonlinear stiffness existed in driving microbeams and sensing microbeams at the same time. Tatar et al. [20] successfully linearized the electrostatic nonlinearity at the driving comb using a formed comb with a tuned cubic hardening compensation in a triple symmetric silicon-on-insulator (SOI)-MEMS micro gyroscope. Ding et al. [32] proposed an improved digital phase-locked driving method based on the comparison of two methods for controlling stiffness nonlinearity under large amplitude. The stability of the control loop was significantly improved at the expense of the larger driving force. Shang et al. [33] studied a 1-DOF micro-gyroscope model driven by parametric excitation with stiffness nonlinearity using analytical and numerical methods. The influence mechanism of system parameters on amplitude-frequency characteristics and bifurcation behavior of driving and sensing modes was revealed. It was found that the variation of excitation frequency has the potential to cause complex dynamic behaviors when the micro gyro vibration system was under 1:1 internal resonance or large carrier angular velocity, such as multi-stable solution, amplitude jump, and almost periodic response.

Parametric excitation for MEMS gyroscopes can provide resonance in both the drive and the sense modes, even with mismatched natural frequencies. Pakniyat et al. [34] studied the requirements for such a condition by analyzing the effect of each factor on the steady state amplitudes of the two modes. By comparing the gyroscope with matched natural frequencies with a gyroscope with mismatching modes, the result showed that parametric excitation was able to provide high accuracy and robustness for MEMS gyroscopes. Min et al. [35] investigated the partial and full chaotic synchronizations of two nonlinear gyroscope systems with/without noise. The study showed that the simple feedback control can make the noised gyroscope system synchronizing with chaotic behaviors of the expected gyroscope system, and a novel synchronization methodology was presented.

To date, the traditional 2-DOF micro gyroscope, which utilizes the 1-DOF drive-mode and sense-mode, is the mainstream of micro gyroscopes nonlinear dynamics. In fact, nonlinear phenomena are more common in micro gyros with MDOF drive-mode and sense-mode, and the nonlinear dynamic behavior is more complicated. With the gradual popularization of MDOF micro gyro applications [36,37,38,39,40,41,42], facing the possible nonlinear dynamic problems in the MDOF micro gyro system is unavoidable. In the MDOF micro gyroscope system, since the displacement of the sense module is much smaller than that of the drive module, the stiffness nonlinearity resulting from the large deformation of the elastic beam in the drive module could become the main nonlinearity source. Therefore, a new type of micro gyroscope system with 2-DOF drive mode and 2-DOF sense mode was proposed in this study, which takes the stiffness nonlinearity of two drive modes into account simultaneously.

In this paper, the gyroscope’s nonlinear dynamic model was developed, which considered the stiffness nonlinearity of two drive modes. The method of multiple scales (MMS) was applied to determine the response of the system under the conditions of primary resonance and 1:1 internal resonance. Then, the Runge-Kutta method was applied to verify the theoretical results. The influence of system parameters on the dynamic characteristics of drive and sense modes were discussed, such as sensitivity, bandwidth, stability, etc. The local bifurcation analysis of the system was carried out. Through the study of the bifurcation behavior in different parameter spaces, the influence of system parameters on the environmental robustness was discussed, which provides theoretical guidance for the design of the MDOF micro gyroscope structure.

## 2. Operational Principle of the Gyroscope

In this paper, a typical 4-DOF micromachinery gyroscope with 2-DOF drive-mode and sense-mode [13] was considered. The schematic diagram of this gyroscope is shown in Figure 1. This gyroscope is mainly composed of driving mass, decoupled frame, proof mass, detection mass, elastic microbeams, and comb electrodes. As shown in Figure 1, the drive direction is along *x* axis, the sense direction is along *y* axis, and *Ω_z_* is the input angular velocity perpendicular to the *x*-*y* plane. The decoupled mass *m_f_* and the proof mass *m*_2_ form a two-stage decoupling structure, which can isolate the drive-mode and the sense-mode. The driving mass *m*_1_ vibrates in *x* direction under the effect of the driving comb electrodes, and the decoupled mass *m_f_* starts to vibrate in *x* direction due to the effect of the microbeam *k*_2_ when the gyroscope works. At the same time, the proof mass *m_f_* vibrates along the *x* direction with the decoupled mass under the effect of the microbeam *k*_4_. Because of the Coriolis effect, the vibration in *x* direction causes the resonance in *y* direction when the system has angular velocity *Ω_z_* input in the vertical direction of the *x-y* plane. Then, the proof mass *m*_2_ and the detection mass *m*_3_ vibrate along the *y* direction under the constraint of the microbeam *k*_4_, *k*_5_, and *k*_6_. The displacement of detection mass *m*_3_ in *y* direction is the detection output of the gyroscope, which increases as the angular velocity increases. The detection output amplitude is proportional to the input angular velocity *Ω_z_* during structural resonance, so the input angular velocity *Ω_z_* of the carrier can be obtained by measuring the output amplitude.

Because the mass of the elastic beam is far less than the vibrating mass, it can be neglected when the micro gyroscope rotates at a constant angular velocity in the *x-y* plane. Therefore, the 4-DOF lumped parameter model can be used to describe the vibration of the micro gyroscope in the *x-y* plane, which is shown in Figure 2.

In addition, the working environment of such a gyroscope is consistent with [13], the air damping is relatively small, and the nonlinear factor of damping can be ignored. Thus, it can be assumed that the damping of the drive and sense directions in the system are linear damping. The dynamic equations of the drive and sense directions of the gyroscope system are established by Figure 2, respectively. The dynamic equations are written as:

Drive direction:(1)$$\{\begin{array}{c} m_{1}{\ddot{x}}_{1}+(c_{1}+c_{2}){\dot{x}}_{1}-c_{2}{\dot{x}}_{2}+(k_{1}+k_{2})x_{1}-k_{2}x_{2}=F_{d}\\ (m_{2}+m_{f}){\ddot{x}}_{2}-c_{2}{\dot{x}}_{1}+(c_{2}+c_{3}){\dot{x}}_{2}-k_{2}x_{1}+(k_{2}+k_{3})x_{2}=0 \end{array},$$

Sense direction:(2)$$\{\begin{array}{c} m_{2}{\ddot{y}}_{1}+(c_{4}+c_{5}){\dot{y}}_{1}-c_{5}{\dot{y}}_{2}+(k_{4}+k_{5})y_{1}-k_{5}y_{2}=F_{c}\\ m_{3}{\ddot{y}}_{2}-c_{5}{\dot{y}}_{1}+(c_{5}+c_{6}){\dot{y}}_{2}-k_{5}y_{1}+(k_{5}+k_{6})y_{2}=0 \end{array},$$
where *m_i_* (*i* = 1, 2, 3, *f*) represents the mass of each mass, *c_i_* (*i* = 1, 2, 3, 4, 5, 6) represents each damping coefficient, *k_i_* (*i* = 1, 2, 3, 4, 5, 6) represents the stiffness coefficient of each elastic microbeam, *x_i_* (*i* = 1, 2) and *y_i_* (*i* = 1, 2) represent the displacement of the *i*-th degree-of-freedom in *x* direction and *y* direction respectively, and *F_d_* and *F_c_* are electrostatic driving force and Coriolis force, respectively. *F_d_* = *F*cos*ω*_0_*t*, $F_{c}=-2m_{2}\Omega_{z}{\dot{x}}_{2}$. *F* is the amplitude of electrostatic driving force, *ω*_0_ is the frequency of electrostatic driving force, and *Ω_z_* is input angular velocity of the gyroscope. The values of these physical parameters [13] are shown in Table 1.

## 3. Perturbation Analysis

In order to consider the case where the model drive beams have nonlinear stiffness, a cubic stiffness term was added to the original drive direction dynamic equations. The nonlinear dynamic equations can be obtained:(3)$$\{\begin{array}{c} m_{1}{\ddot{x}}_{1}+(c_{1}+c_{2}){\dot{x}}_{1}-c_{2}{\dot{x}}_{2}+(k_{1}+k_{2})x_{1}-k_{2}x_{2}+K_{1}x_{1}^{3}=F\cos\omega_{0}t\\ (m_{2}+m_{f}){\ddot{x}}_{2}-c_{2}{\dot{x}}_{1}+(c_{2}+c_{3}){\dot{x}}_{2}-k_{2}x_{1}+(k_{2}+k_{3})x_{2}+K_{2}x_{2}^{3}=0 \end{array}$$
where *K_i_* (*i* = 1, 2) represents the stiffness nonlinear coefficient of the *i*-th degree of freedom in the drive direction. Simplify Equation (3), and the following equation can be obtained:(4)$$\{\begin{array}{c} {\ddot{x}}_{1}+\alpha_{3}{\dot{x}}_{1}-\alpha_{4}{\dot{x}}_{2}+\alpha_{1}x_{1}-\alpha_{2}x_{2}+\alpha_{5}x_{1}^{3}=F^{\prime }\cos\omega_{0}t\\ {\ddot{x}}_{2}-\beta_{3}{\dot{x}}_{1}+\beta_{4}{\dot{x}}_{2}-\beta_{1}x_{1}+\beta_{2}x_{2}+\beta_{5}x_{2}^{3}=0 \end{array}$$
where $$\alpha_{1}=\frac{k_{1}+k_{2}}{m_{1}},\alpha_{2}=\frac{k_{2}}{m_{1}},\alpha_{3}=\frac{c_{1}+c_{2}}{m_{1}},\alpha_{4}=\frac{c_{2}}{m_{1}},\alpha_{5}=\frac{K_{1}}{m_{1}},F^{\prime }=\frac{F}{m_{1}},$$
$$\beta_{1}=\frac{k_{2}}{m_{2}+m_{f}},\beta_{2}=\frac{k_{2}+k_{3}}{m_{2}+m_{f}},\beta_{3}=\frac{c_{2}}{m_{2}+m_{f}},\beta_{4}=\frac{c_{2}+c_{3}}{m_{2}+m_{f}},\beta_{5}=\frac{K_{2}}{m_{2}+m_{f}}.$$

Equation (4) is the forced vibration of Duffing system with damping under harmonic excitation. The MMS is used to investigate the approximate periodic response of Equation (4). In order to study the dynamic behavior of the system near the internal resonance, the primary resonance and 1:1 internal resonance of the system were considered. To describe the nearness of the resonance, two detuning parameters *σ*_1_ and *σ*_2_ were introduced and defined by:(5)$$\omega_{0}^{2}=\omega_{1}^{2}+\varepsilon^{2}\sigma_{1},\;\omega_{2}^{2}=\omega_{1}^{2}+\varepsilon^{2}\sigma_{2},$$
where *ω*_1_ and *ω*_2_ are the resonance frequencies of the first-order mode and the second-order mode in the drive direction, respectively. *ε* is introduced as a small nondimensional bookkeeping parameter.

Since the primary resonance of *ω*_0_ close to *ω*_1_ was studied, the electrostatic force term $F^{\prime }=O(\varepsilon^{3})$ and the damping term $c_{i}=O(\varepsilon^{2})$ were considered here for making the effects of the damping term and the forcing term appear in the same perturbation equation as the nonlinear effect. Substituting Equation (5) into Equation (4) and scaling the dissipative terms, Equation (4) can be modified as:(6)$$\{\begin{array}{c} {\ddot{x}}_{1}+\omega_{0}^{2}x_{1}=-\varepsilon^{2}{\hat{\alpha }}_{1}x_{1}+\varepsilon^{2}{\hat{\alpha }}_{2}x_{2}-\varepsilon^{2}{\hat{\alpha }}_{3}{\dot{x}}_{1}+\varepsilon^{2}{\hat{\alpha }}_{4}{\dot{x}}_{2}-\alpha_{5}x_{1}^{3}+\varepsilon^{3}f\cos\omega_{0}t+\varepsilon^{2}{\hat{\omega }}_{1}^{2}x_{1}+\varepsilon^{2}\sigma_{1}x_{1}\\ {\ddot{x}}_{2}+\omega_{0}^{2}x_{2}=\varepsilon^{2}{\hat{\beta }}_{1}x_{1}-\varepsilon^{2}{\hat{\beta }}_{2}x_{2}+\varepsilon^{2}{\hat{\beta }}_{3}{\dot{x}}_{1}-\varepsilon^{2}{\hat{\beta }}_{4}{\dot{x}}_{2}-\beta_{5}x_{2}^{3}+\varepsilon^{2}{\hat{\omega }}_{2}^{2}x_{2}+\varepsilon^{2}(\sigma_{1}-\sigma_{2})x_{2} \end{array}$$
where $\alpha_{i}=\varepsilon^{2}{\hat{\alpha }}_{i}\;(i=1,2,3,4)$, $\beta_{i}=\varepsilon^{2}{\hat{\beta }}_{i}\;(i=1,2,3,4)$, $\omega_{i}^{2}=\varepsilon^{2}{\hat{\omega }}_{i}^{2}\;(i=1,2)$, $F^{\prime }=\varepsilon^{3}f$.

The approximate solution of Equation (6) can be written in the following form:(7)$$\begin{array}{l} x_{1}=\varepsilon x_{11}(T_{0},T_{2})+\varepsilon^{3}x_{13}(T_{0},T_{2})\\ x_{2}=\varepsilon x_{21}(T_{0},T_{2})+\varepsilon^{3}x_{23}(T_{0},T_{2}) \end{array}$$
where $T_{n}=\varepsilon^{n}t,\;(n=0,1,2)$.

Substituting Equation (7) into Equation (6) and equating the coefficients of like powers of *ε*, the following partial differential equations can be obtained:(8)$$\begin{array}{ll} O(\varepsilon^{1}): & D_{0}^{2}x_{11}+\omega_{0}^{2}x_{11}=0\\ & D_{0}^{2}x_{21}+\omega_{0}^{2}x_{21}=0 \end{array}$$
(9)$$\begin{array}{lll} O(\varepsilon^{3}):\; & D_{0}^{2}x_{13}+\omega_{0}^{2}x_{13}= & -D_{0}(2D_{2}x_{11}+{\hat{\alpha }}_{3}x_{11})-{\hat{\alpha }}_{1}x_{11}+{\hat{\alpha }}_{2}x_{21}+{\hat{\alpha }}_{4}D_{0}x_{21}\\ & & -\alpha_{5}x_{11}^{3}+f\cos\omega_{0}t+{\hat{\omega }}_{1}^{2}x_{11}+\sigma_{1}x_{11}\\ & D_{0}^{2}x_{23}+\omega_{0}^{2}x_{23}= & -D_{0}(2D_{2}x_{21}+{\hat{\beta }}_{4}x_{21})+{\hat{\beta }}_{1}x_{11}-{\hat{\beta }}_{2}x_{21}+{\hat{\beta }}_{3}D_{0}x_{11}\\ & & -\beta_{5}x_{21}^{3}+{\hat{\omega }}_{2}^{2}x_{21}+(\sigma_{1}-\sigma_{2})x_{21} \end{array}$$
where $D_{n}=\frac{\partial }{\partial T_{n}},n=(0,1,2)$.

The general solution of Equation (8) can be written as:(10)$$\begin{array}{l} x_{11}(T_{0},T_{2})=A_{11}(T_{2})\exp(i\omega_{0}T_{0})+{\bar{A}}_{11}(T_{2})\exp(-i\omega_{0}T_{0})\\ x_{21}(T_{0},T_{2})=A_{21}(T_{2})\exp(i\omega_{0}T_{0})+{\bar{A}}_{21}(T_{2})\exp(-i\omega_{0}T_{0}) \end{array}$$

Here, it is convenient to express *A*_11_ and *A*_21_ in the polar form:(11)$$\begin{array}{l} A_{11}(T_{2})=\frac{1}{2}a_{1}(T_{2})\exp\left[i\theta_{1}(T_{2})\right]\\ A_{21}(T_{2})=\frac{1}{2}a_{2}(T_{2})\exp\left[i\theta_{2}(T_{2})\right] \end{array}$$
where *a*_1_ and *a*_2_ indicate the amplitudes of the first-order vibration mode and the second-order vibration mode in the drive direction, respectively. *θ*_1_ and *θ*_2_ indicate the initial phases of the first-order vibration mode and the second-order vibration mode in the drive direction, respectively.

Next, we substituted Equations (10) and (11) into Equation (9). According to the solvability condition without secular terms, the average equations of amplitudes and phases can be obtained:(12)$$\begin{array}{ll} {\dot{a}}_{1}= & -\frac{{\hat{\alpha }}_{3}a_{1}}{2}+\frac{{\hat{\alpha }}_{2}a_{2}}{2\omega_{0}}\sin\phi +\frac{{\hat{\alpha }}_{4}a_{2}}{2}\cos\phi -\frac{f}{2\omega_{0}}\sin\theta_{1}\\ {\dot{\theta }}_{1}= & \frac{{\hat{\alpha }}_{1}-{\hat{\omega }}_{1}^{2}-\sigma_{1}}{2\omega_{0}}-\frac{{\hat{\alpha }}_{2}a_{2}}{2\omega_{0}a_{1}}\cos\phi +\frac{{\hat{\alpha }}_{4}a_{2}}{2a_{1}}\sin\phi +\frac{3\alpha_{5}a_{1}^{2}}{8\omega_{0}}-\frac{f}{2\omega_{0}a_{1}}\cos\theta_{1}\\ {\dot{a}}_{2}= & -\frac{{\hat{\beta }}_{4}a_{2}}{2}-\frac{{\hat{\beta }}_{1}a_{1}}{2\omega_{0}}\sin\phi +\frac{{\hat{\beta }}_{3}a_{1}}{2}\cos\phi \\ \dot{\phi }= & -\frac{{\hat{\beta }}_{1}a_{1}}{2\omega_{0}a_{2}}\cos\phi +\frac{{\hat{\beta }}_{2}-{\hat{\omega }}_{2}^{2}+\sigma_{2}-\sigma_{1}}{2\omega_{0}}-\frac{{\hat{\beta }}_{3}a_{1}}{2a_{2}}\sin\phi +\frac{3\beta_{5}a_{2}^{2}}{8\omega_{0}}-\\ & \frac{{\hat{\alpha }}_{1}-{\hat{\omega }}_{1}^{2}-\sigma_{1}}{2\omega_{0}}+\frac{{\hat{\alpha }}_{2}a_{2}}{2\omega_{0}a_{1}}\cos\phi -\frac{{\hat{\alpha }}_{4}a_{2}}{2a_{1}}\sin\phi -\frac{3\alpha_{5}a_{1}^{2}}{8\omega_{0}}+\frac{f}{2\omega_{0}a_{1}}\cos\theta_{1} \end{array}$$
where $\phi =\theta_{2}-\theta_{1}$.

To determine the stability of the periodic solution, the Jacobian matrix of Equation (12) at (*a*_10_, *θ*_10_, *a*_20_, *ϕ*_0_) is given:(13)$$J=\left[\begin{array}{cccc} J_{11} & J_{12} & J_{13} & J_{14}\\ J_{21} & J_{22} & J_{23} & J_{24}\\ J_{31} & J_{32} & J_{33} & J_{34}\\ J_{41} & J_{42} & J_{43} & J_{44} \end{array}\right].$$where 

$J_{11}=-\frac{{\hat{\alpha }}_{3}}{2}$, $J_{12}=-\frac{f}{2\omega_{0}}\cos\theta_{10}$, $J_{13}=\frac{{\hat{\alpha }}_{2}}{2\omega_{0}}\sin\varphi_{0}+\frac{{\hat{\alpha }}_{4}}{2}\cos\varphi_{0},\;$$J_{14}=\frac{{\hat{\alpha }}_{2}a_{20}}{2\omega_{0}}\cos\varphi_{0}-\frac{{\hat{\alpha }}_{4}a_{20}}{2}\sin\varphi_{0}$, $J_{21}=\frac{{\hat{\alpha }}_{2}a_{20}}{2\omega_{0}a_{10}^{2}}\cos\varphi_{0}-\frac{{\hat{\alpha }}_{4}a_{20}}{2a_{10}^{2}}\sin\varphi_{0}+\frac{3\alpha_{5}a_{10}}{4\omega_{0}}+\frac{f}{2\omega_{0}a_{10}^{2}}\cos\theta_{10}$, $J_{22}=\frac{f}{2\omega_{0}a_{10}}\sin\theta_{10}$, $J_{23}=-\frac{{\hat{\alpha }}_{2}}{2\omega_{0}a_{10}}\cos\varphi_{0}+\frac{{\hat{\alpha }}_{4}}{2a_{10}}\sin\varphi_{0}$, $J_{24}=\frac{{\hat{\alpha }}_{2}a_{20}}{2\omega_{0}a_{10}}\sin\varphi_{0}+\frac{{\hat{\alpha }}_{4}a_{20}}{2a_{10}}\cos\varphi_{0}$, $J_{31}=-\frac{{\hat{\beta }}_{1}}{2\omega_{0}}\sin\varphi_{0}+\frac{{\hat{\beta }}_{3}}{2}\cos\varphi_{0}$, 

$J_{32}=0$, $J_{33}=-\frac{{\hat{\beta }}_{4}}{2}$, $J_{34}=-\frac{{\hat{\beta }}_{1}a_{10}}{2\omega_{0}}\cos\varphi_{0}-\frac{{\hat{\beta }}_{3}a_{10}}{2}\sin\varphi_{0}$, $J_{41}=-\frac{{\hat{\beta }}_{1}}{2\omega_{0}a_{20}}\cos\varphi_{0}-\frac{{\hat{\beta }}_{3}}{2a_{20}}\sin\varphi_{0}-\frac{{\hat{\alpha }}_{2}a_{20}}{2\omega_{0}a_{10}^{2}}\cos\varphi_{0}+\frac{{\hat{\alpha }}_{4}a_{20}}{2a_{10}^{2}}\sin\varphi_{0}-\frac{3\alpha_{5}a_{10}}{4\omega_{0}}-\frac{f}{2\omega_{0}a_{10}^{2}}\cos\theta_{10}$, $J_{42}=-\frac{f}{2\omega_{0}a_{10}}\sin\theta_{10}$, $J_{43}=\frac{{\hat{\beta }}_{1}a_{10}}{2\omega_{0}a_{20}^{2}}\cos\varphi_{0}+\frac{{\hat{\beta }}_{3}a_{10}}{2a_{20}^{2}}\sin\varphi_{0}+\frac{3\beta_{5}a_{20}}{4\omega_{0}}+\frac{{\hat{\alpha }}_{2}}{2\omega_{0}a_{10}}\cos\varphi_{0}-\frac{{\hat{\alpha }}_{4}}{2a_{10}}\sin\varphi_{0}$, $J_{44}=\frac{{\hat{\beta }}_{1}a_{10}}{2\omega_{0}a_{20}}\sin\varphi_{0}-\frac{{\hat{\beta }}_{3}a_{10}}{2a_{20}}\cos\varphi_{0}-\frac{{\hat{\alpha }}_{2}a_{20}}{2\omega_{0}a_{10}}\sin\varphi_{0}-\frac{{\hat{\alpha }}_{4}a_{20}}{2a_{10}}\cos\varphi_{0}$.

The system is stable when all the matrix eigenvalues are negative. Otherwise, the system is unstable.

In Equation (12), by imposing the conditions: ${\dot{a}}_{1}=0\;,\;{\dot{\theta }}_{1}=0\;,\;{\dot{a}}_{2}=0\;,\;\dot{\phi }=0$, the frequency response equations about *a*_1_ and *a*_2_ can be derived as follows:(14)$$\begin{array}{l} a_{2}^{2}{\left[-4{\hat{\beta }}_{1}{\hat{\beta }}_{2}-3{\hat{\beta }}_{1}\beta_{5}a_{2}^{2}+4{\hat{\beta }}_{1}(\sigma_{1}-\sigma_{2})-4{\hat{\beta }}_{3}{\hat{\beta }}_{4}\omega_{0}^{2}+4{\hat{\beta }}_{1}{\hat{\omega }}_{2}^{2}\right]}^{2}+\\ a_{2}^{2}\omega_{0}^{2}{\left[4{\hat{\beta }}_{2}{\hat{\beta }}_{3}-4{\hat{\beta }}_{1}{\hat{\beta }}_{4}+3{\hat{\beta }}_{3}\beta_{5}a_{2}^{2}-4{\hat{\beta }}_{3}(\sigma_{1}-\sigma_{2})-4{\hat{\beta }}_{3}{\hat{\omega }}_{2}^{2}\right]}^{2}=16a_{1}^{2}{\left({\hat{\beta }}_{1}^{2}+{\hat{\beta }}_{3}^{2}\omega_{0}^{2}\right)}^{2} \end{array}$$
(15)$$\begin{array}{l} \omega_{0}^{2}[4{\hat{\alpha }}_{3}{\hat{\beta }}_{1}^{2}a_{1}^{2}-4{\hat{\alpha }}_{4}{\hat{\beta }}_{1}{\hat{\beta }}_{2}a_{2}^{2}-4{\hat{\alpha }}_{2}{\hat{\beta }}_{2}{\hat{\beta }}_{3}a_{2}^{2}+4{\hat{\alpha }}_{2}{\hat{\beta }}_{1}{\hat{\beta }}_{4}a_{2}^{2}-3{\hat{\alpha }}_{4}{\hat{\beta }}_{1}\beta_{5}a_{2}^{4}-3{\hat{\alpha }}_{2}{\hat{\beta }}_{3}\beta_{5}a_{2}^{4}\\ +4{\hat{\alpha }}_{4}{\hat{\beta }}_{1}a_{2}^{2}(\sigma_{1}-\sigma_{2})+4{\hat{\alpha }}_{2}{\hat{\beta }}_{3}a_{2}^{2}(\sigma_{1}-\sigma_{2})+4{\hat{\alpha }}_{3}{\hat{\beta }}_{3}^{2}\omega_{0}^{2}a_{1}^{2}-4{\hat{\alpha }}_{4}{\hat{\beta }}_{3}{\hat{\beta }}_{4}\omega_{0}^{2}a_{2}^{2}+4{\hat{\alpha }}_{4}{\hat{\beta }}_{1}{\hat{\omega }}_{2}^{2}a_{2}^{2}\\ +4{\hat{\alpha }}_{2}{\hat{\beta }}_{3}{\hat{\omega }}_{2}^{2}a_{2}^{2}{]}^{2}+[4{\hat{\alpha }}_{1}{\hat{\beta }}_{1}^{2}a_{1}^{2}+3\alpha_{5}{\hat{\beta }}_{1}^{2}a_{1}^{4}-4{\hat{\alpha }}_{2}{\hat{\beta }}_{1}{\hat{\beta }}_{2}a_{2}^{2}-3{\hat{\alpha }}_{2}{\hat{\beta }}_{1}\beta_{5}a_{2}^{4}+4{\hat{\alpha }}_{2}{\hat{\beta }}_{1}a_{2}^{2}(\sigma_{1}-\sigma_{2})\\ -4{\hat{\beta }}_{1}^{2}a_{1}^{2}\sigma_{1}+4{\hat{\alpha }}_{4}{\hat{\beta }}_{2}{\hat{\beta }}_{3}\omega_{0}^{2}a_{2}^{2}+4{\hat{\alpha }}_{1}{\hat{\beta }}_{3}^{2}\omega_{0}^{2}a_{1}^{2}+3\alpha_{5}{\hat{\beta }}_{3}^{2}\omega_{0}^{2}a_{1}^{4}-4{\hat{\alpha }}_{4}{\hat{\beta }}_{1}{\hat{\beta }}_{4}\omega_{0}^{2}a_{2}^{2}-4{\hat{\alpha }}_{2}{\hat{\beta }}_{3}{\hat{\beta }}_{4}\omega_{0}^{2}a_{2}^{2}\\ +3{\hat{\alpha }}_{4}{\hat{\beta }}_{3}{\hat{\beta }}_{5}\omega_{0}^{2}a_{2}^{4}-4{\hat{\alpha }}_{4}{\hat{\beta }}_{3}\omega_{0}^{2}a_{2}^{2}(\sigma_{1}-\sigma_{2})-4{\hat{\beta }}_{3}^{2}\omega_{0}^{2}a_{1}^{2}\sigma_{1}-4{\hat{\beta }}_{1}^{2}{\hat{\omega }}_{1}^{2}a_{1}^{2}-4{\hat{\beta }}_{3}^{2}\omega_{0}^{2}{\hat{\omega }}_{1}^{2}a_{1}^{2}+4{\hat{\alpha }}_{2}{\hat{\beta }}_{1}{\hat{\omega }}_{2}^{2}a_{2}^{2}\\ -4{\hat{\alpha }}_{4}{\hat{\beta }}_{3}\omega_{0}^{2}{\hat{\omega }}_{2}^{2}a_{2}^{2}{]}^{2}=16a_{1}^{2}f^{2}{\left({\hat{\beta }}_{1}^{2}+{\hat{\beta }}_{3}^{2}\omega_{0}^{2}\right)}^{2} \end{array}$$

Using the Newton iteration method and pseudo-arc length method to solve nonlinear coupled Equations (14) and (15), and amplitude frequency response of *a*_1_ and *a*_2_ can be obtained. By discussing the Equations (14) and (15), the influence of the system parameters on the amplitude of the drive response can be obtained.

To obtain the detection output response under the driving nonlinear stiffness, the complex exponential method (CEM) was used to solve the dynamic equation in the sense direction. Equation (2) can be expressed as:(16)$$M\ddot{y}+C\dot{y}+Ky=F_{c}(t)$$
where *M* is the mass matrix of micro gyro in sense direction, *C* is the damping matrix, *K* is the stiffness matrix, and *F_c_*(*t*) is the column vector of the excitation force. According to condition $\left|K-\omega^{2}M\right|=0$, the resonant frequency of first and second modes in sense direction *ω*_3_ and *ω*_4_ can be sought out. Here,
$$M=\left[\begin{array}{cc} m_{2} & 0\\ 0 & m_{3} \end{array}\right],\;C=\left[\begin{array}{cc} c_{4}+c_{5} & -c_{5}\\ -c_{5} & c_{5}+c_{6} \end{array}\right],\;K=\left[\begin{array}{cc} k_{4}+k_{5} & -k_{5}\\ -k_{5} & k_{5}+k_{6} \end{array}\right],\;F_{c}(t)=\left[\begin{array}{c} -2m_{2}\Omega_{z}{\dot{x}}_{2}\\ 0 \end{array}\right]$$

From the previous calculation, the approximate solution of *x*_2_ is *a*_2_cos(*ω*_0_*t*−*θ*_2_). Therefore, the Coriolis force can be expressed as 2*m*_2_*Ω_z_a*_2_*ω*_0_sin(*ω*_0_*t*−*θ*_2_). *f_c_* is introduced as the amplitude of Coriolis force, so *f_c_* = 2*m*_2_*Ω_z_a*_2_*ω*_0_. Obviously, the amplitude of Coriolis force is related to the drive excitation frequency. The CEM was used to solve and determine the steady-state response of the system. The steady-state response of Equation (2) can be written as:(17)$$y_{1}(t)={\bar{b}}_{1}e^{j(\omega_{0}t-\theta_{2})}\;,\;y_{2}(t)={\bar{b}}_{2}e^{j(\omega_{0}t-\theta_{2})}$$
where ${\bar{b}}_{1}\;,\;{\bar{b}}_{2}$ are complex amplitudes.

Substituting Equation (16) into Equation (2), the following can be obtained:(18)$$\begin{array}{ll} \left[\begin{array}{c} {\bar{b}}_{1}\\ {\bar{b}}_{2} \end{array}\right] & ={\left[\begin{array}{cc} k_{4}+k_{5}-\omega_{0}^{2}m_{2}+j\omega_{0}(c_{4}+c_{5}) & -k_{5}-j\omega_{0}c_{5}\\ -k_{5}-j\omega_{0}c_{5} & k_{5}+k_{6}-\omega_{0}^{2}m_{3}+j\omega_{0}(c_{5}+c_{6}) \end{array}\right]}^{-1}\left[\begin{array}{c} f_{c}\\ 0 \end{array}\right]\\ & =\frac{f_{c}}{\nabla (\omega_{0})}\left[\begin{array}{c} k_{5}+k_{6}-\omega_{0}^{2}m_{3}+j\omega_{0}(c_{5}+c_{6})\\ k_{5}+j\omega_{0}c_{5} \end{array}\right] \end{array}$$
where $\nabla (\omega_{0})=\left[k_{4}+k_{5}-\omega_{0}^{2}m_{2}+j\omega_{0}(c_{4}+c_{5})\right]\left[k_{5}+k_{6}-\omega_{0}^{2}m_{3}+j\omega_{0}(c_{5}+c_{6})\right]-{\left(-k_{5}-j\omega_{0}c_{5}\right)}^{2}$.

Therefore, the complex amplitude can be obtained as:(19)$$\{\begin{array}{c} {\bar{b}}_{1}=\frac{k_{5}+k_{6}-\omega_{0}^{2}m_{3}+j\omega_{0}(c_{5}+c_{6})}{\nabla (\omega_{0})}f_{c}=b_{1}e^{-\vartheta_{1}}\\ {\bar{b}}_{2}=\frac{k_{5}+j\omega_{0}c_{5}}{\nabla (\omega_{0})}f_{c}=b_{2}e^{-\vartheta_{2}} \end{array}$$
where *b*_1_, *b*_2_ and *ϑ*_1_, *ϑ*_2_ are the amplitudes and initial phases of the sense modal steady-state response, respectively.

Finally, the steady-state amplitudes in sense direction can be derived as:(20)$$\{\begin{array}{c} b_{1}=\frac{2m_{2}\Omega_{z}a_{2}\omega_{0}{\left[{\left(c_{5}+c_{6}\right)}^{2}\omega_{0}^{2}+{\left(k_{5}+k_{6}-m_{3}\omega_{0}^{2}\right)}^{2}\right]}^{\frac{1}{2}}}{{\left\{\begin{array}{l} {\left[-2c_{5}k_{5}\omega_{0}+(c_{5}+c_{6})(k_{4}+k_{5}-m_{2}\omega_{0}^{2})\omega_{0}+(c_{4}+c_{5})(k_{5}+k_{6}-m_{3}\omega_{0}^{2})\omega_{0}\right]}^{2}\\ +{\left[-k_{5}^{2}+c_{5}^{2}\omega_{0}^{2}-(c_{4}+c_{5})(c_{5}+c_{6})\omega_{0}^{2}+(k_{4}+k_{5}-m_{2}\omega_{0}^{2})(k_{5}+k_{6}-m_{3}\omega_{0}^{2})\right]}^{2} \end{array}\right\}}^{\frac{1}{2}}}\\ b_{2}=\frac{2m_{2}\Omega_{z}a_{2}\omega_{0}{\left(k_{5}^{2}+c_{5}^{2}\omega_{0}^{2}\right)}^{\frac{1}{2}}}{{\left\{\begin{array}{l} {\left[-2c_{5}k_{5}\omega_{0}+(c_{5}+c_{6})(k_{4}+k_{5}-m_{2}\omega_{0}^{2})\omega_{0}+(c_{4}+c_{5})(k_{5}+k_{6}-m_{3}\omega_{0}^{2})\omega_{0}\right]}^{2}\\ +{\left[-k_{5}^{2}+c_{5}^{2}\omega_{0}^{2}-(c_{4}+c_{5})(c_{5}+c_{6})\omega_{0}^{2}+(k_{4}+k_{5}-m_{2}\omega_{0}^{2})(k_{5}+k_{6}-m_{3}\omega_{0}^{2})\right]}^{2} \end{array}\right\}}^{\frac{1}{2}}} \end{array}$$

### 3.1. Stiffness Nonlinear Analysis

The calculation parameters were selected as follows: Excitation force amplitude *F* = 3 × 10^−5^ N, *ε* = 1. From the parameters in Table 1, the natural frequencies of the first-order and second-order drive modes can be obtained as *ω*_1_ = 30904.11 rad/s, *ω*_2_ = 31880.86 rad/s. In the case of *σ*_2_ = 5.91 × 10^7^, the effect of internal resonance parameter changes on micro gyroscope response were not considered in this section. The amplitude-frequency response curves of the first-order and second-order drive modes with different nonlinear stiffness coefficients are shown in Figure 3 and Figure 4. In order to verify the theoretical analytical solution obtained by MMS, the Runge-Kutta numerical method was used to solve Equation (1). Comparison to the theoretical solution, there is a good agreement.

As shown in Figure 3, the response of the first-order drive mode was basically the same as that of the linear system, when the nonlinear stiffness coefficients *K*_1_ and *K*_2_ are less than 10^11^ N/m^3^. In this case, the influence of stiffness nonlinearity was negligible. The obvious nonlinear hardening characteristics began to appear at the second peak value, when the nonlinear stiffness coefficients *K*_1_ and *K*_2_ reached 10^11.5^ N/m^3^. The sensitivity of the first peak decreased slightly, but the second peak increased considerably. The nonlinear elastic force was only 2.24% of the linear elastic force in this case. Typical nonlinear characteristics, such as multiple steady state solution, amplitude jump, and frequency offset appeared at the second peak, when the nonlinear stiffness coefficients *K*_1_ and *K*_2_ reached 10^12^ N/m^3^. In comparison to the case of the linear stiffness, the sensitivity at the first peak was reduced by 25.5%, and the first-order resonance frequency was slightly shifted to the right. The sensitivity at the second peak was increased by 77.8% and the second-order resonance frequency was obviously shifted to the right, but the sensitivity was reduced by 40% at the original resonance frequency. At this time, the nonlinear elastic force was 5.42% of the linear elastic force. The sensitivity stability near the second resonance frequency was destroyed, and there was another stable solution far below the peak value at the instantaneous natural frequency corresponding to the second peak value, because of the amplitude jumping behavior and the appearance of multiple steady state solutions. This phenomenon was due to the dependence of the nonlinear system on the initial conditions. The system was a periodic motion of smaller amplitude when the operating frequency was lowered from the high frequency to the resonant frequency, while the system was a periodic motion of larger amplitude when the operating frequency rose from the low frequency to the resonant frequency. As shown in Figure 3d, the dashed line was an unstable region of the approximate periodic solution and the intermediate solution branch of the multiple steady state solution. The motion corresponding to this area is unlikely to occur in a real micro gyroscope system.

The response of second-order drive mode is shown in Figure 4. The amplitude-frequency response was basically the same as that under linear stiffness, when the nonlinear stiffness coefficients *K*_1_ and *K*_2_ were less than 10^11^ N/m^3^. The influence of stiffness nonlinearity was negligible in this case. The amplitude-frequency response began to appear hardened and the sensitivity was basically unchanged. The second-order resonant frequency was slightly offset to the right when the nonlinear stiffness coefficients *K*_1_ and *K*_2_ reached 10^11.5^ N/m^3^. The response curves showed obvious nonlinear characteristics when the nonlinear stiffness coefficients *K*_1_ and *K*_2_ reached 10^12^ N/m^3^. Nonlinear behaviors, such as multiple steady state solutions and amplitude jump, appeared near the second formant. The *BW* area in Figure 4 represents the 3dB bandwidth of the drive output response. With the increase of nonlinear stiffness, the output response bandwidth remained basically unchanged, but the frequency corresponding to the bandwidth shifted slightly to the right due to the nonlinear frequency offset effect. Within the bandwidth range, the sensitivity of the output response decreased slightly with nonlinear enhancement. However, it is important that the multiple steady state solution region caused by the hardening of the stiffness nonlinearity was outside the bandwidth range, as the system instability caused by the amplitude jump did not occur in the bandwidth range. This is because that the 2-DOF drive-mode structure of the micro-gyro model had a relatively flat and high sensitivity between the two peaks in the amplitude-frequency curve, which made the drive module have high robustness in the working process. In addition, the drive module was not easily affected in the bandwidth region.

Combining Equations (14) and (15), and (20), the relationship between the steady-state amplitude of the sense mode and the driving frequency can be obtained, that is, the amplitude frequency response relationship of the sense mode. The mathematical analysis software *Mathematica* was used to plot the amplitude-frequency response curve of the primary resonance of the first-order and second-order sense modes, as shown in Figure 5 and Figure 6. With Runge-Kutta method used to solve Equations (1) and (2), a series of numerical solutions which coincide with the theoretical analytical solution were obtained, and the correctness of theoretical solution was verified.

As illustrated in Figure 5 and Figure 6, the amplitude frequency response of the first-order and second-order sense modes were similar to that of the drive modes. The response curve was basically the same as that of linear stiffness case when the nonlinear stiffness coefficients are less than 10^11^ N/m^3^. The amplitude-frequency response began to show nonlinear characteristics, and a slight increase would cause a significant change in the shape of the response curve when the stiffness nonlinearity coefficients exceeded 10^11^ N/m^3^. As shown in Figure 5c,d and Figure 6c,d, the steady-state response had typical nonlinear behaviors, such as multiple steady state solution, amplitude jump, and frequency offset. The nonlinear influence was stronger, which can be seen from the bending degree of the peak. The difference is that when the stiffness nonlinearity coefficients increased from 10^11.5^ N/m^3^ to 10^12^ N/m^3^, the frequency band range corresponding to the unstable solution (dashed line) of the first-order sense mode increased, while second-order sense mode decreased. The reduction of the corresponding frequency band range of the unstable solution means that the probability of amplitude jump in steady-state response decreased, that is to say, the possibility of an abrupt change of output signal in the micro-gyroscope system was reduced. As shown in Figure 6, the area marked by *BW* is the bandwidth for sensing the output response, and it is also the detection bandwidth of the micro gyroscope. It can be seen from Figure 6 that the bandwidth was slightly shifted to the right when the nonlinearity was strengthened. The bandwidth was narrowed and the detection sensitivity decreased in the bandwidth range when the nonlinearity was relatively strong. On the other hand, the region of the multi-stable solution was outside the detection bandwidth and did not affect the stability in the bandwidth range because of the nonlinear stiffness hardening. It is similar to the drive output response. Therefore, the instability behavior, such as amplitude jump, will not occur when the micro gyroscope works in the detection bandwidth, even if there is a strong nonlinear stiffness. This is also due to the high robustness of the sense module. The amplitude in the bandwidth is relatively flat and the stiffness nonlinearity has less effect on the inner part of the bandwidth. In addition, the frequency band region of the drive output response bandwidth is highly matched with the sense response bandwidth region, so that the micro gyroscope has a strong resistance to nonlinear factors when operating in the working bandwidth.

### 3.2. System Parameters Analysis

The parameter *σ*_2_ is an internal resonance tuning parameter, which characterizes the proximity of the resonant frequencies of the first-order and second-order drive modes. $\sigma_{2}=\omega_{2}^{2}-\omega_{1}^{2}$, when *ε* = 1. Therefore, both the internal resonance tuning parameters *σ*_2_ and the steady-state response of the system will change when a certain order drive mode resonance frequency shifts under the influence of external conditions. The variation curves of the response amplitude of each degree of freedom of the system are shown in Figure 7 with internal resonance parameter *σ*_2_ at different operating frequency *ω*_0_.

As shown in Figure 7, the operating frequency *ω*_0_ takes three different values (30940.11 rad/s, 31500 rad/s, 32000 rad/s), corresponding to three different values of *σ*_1_ (0, 3.5 × 10^7^, 6.67 × 10^7^), which represent the complete tuning, small detuning, and detuning of the primary resonance, respectively. In Figure 7a, with *σ*_2_ increases the response amplitude of the first-order drive mode decreased from the stable value to the minimum value, and then slowly increased to another stability value, when *σ*_1_ takes different values. However, in Figure 7b, the response amplitude of second-order drive mode increased first, and then decreased gradually after reaching the maximum value with the increase of *σ*_2_. This is because that *ω*_1_ was closer to *ω*_2_, and the internal resonance was closer to the tuning state when *σ*_2_ was relatively small. The partial energy generated by the first-order drive mode primary resonance was transferred to the second-order drive mode, and the response amplitude of the second-order drive mode increased gradually, while the amplitude of the first-order drive mode decreased. The degree of internal resonance detuning became larger, and energy transfer was relatively less with the larger *σ*_2_. The response amplitude of the second-order drive mode gradually decreased, but the first-order drive mode slowly increased.

As shown in Figure 7c,d, in the case of three different *σ*_1_, the response amplitudes of the first-order and second-order sense modes increased first and then decreased with the increase of *σ*_2_. Its change trend is similar to that of the second-order drive mode in Figure 7b. This is because the response amplitude of the second-order drive mode in this system was the output response of the drive direction, which determined the amplitude of the Coriolis force in the sense direction. There is no nonlinear stiffness in the sense direction. The steady-state response amplitudes of the first-order and second-order sense modes are proportional to the Coriolis force amplitude, and the change trend is consistent with that of the second-order drive mode. During the operation of the micro gyroscope, the stiffness of elastic micro-beams decreased because of the softening effect of the elastic modulus when the external temperature rose. Then, the modal resonance frequency was reduced and the drift was generated. Frequency drift results in the change of internal resonance parameters, even at constant excitation frequency. The response amplitude will change or even change greatly. In addition, the stiffness of the elastic micro-beams will be affected when there are errors(deviations) in the processing of them. It will cause the resonance frequency to drift, as well as have a greater impact on the amplitude of the output response. Therefore, the external working conditions and processing errors may have a large impact on the output of the gyroscope system with 1:1 internal resonance. In this case, the error factors should be fully considered, and corresponding error compensation should be made.

In order to study the influence of external input energy on the driving response of gyroscope system more clearly, AC voltage with different amplitudes was applied to both ends of driving electrodes to subject the driving mass to the electrostatic driving force of different amplitudes. In addition, the value of parameter *σ*_2_ was selected in the region where the amplitude existed in the multiple steady-state solutions in Figure 7. Here, *σ*_2_ = 10 × 10^7^, *K*_1_ = *K*_2_ = 10^12^ N/m^3^ were selected. The relationship between the steady-state response amplitude of the first-order and second-order drive modes and electrostatic driving force amplitude were studied under the condition of primary resonance complete tuning (*σ*_1_ = 0) and detuning (*σ*_1_ = 6.67 × 10^7^), as shown in Figure 8.

It can be seen from Figure 8a that both the response amplitudes of first-order and second-order drive modes were uniquely determined. The driving mass and the decoupled frame were both stable periodic motions, under the condition of primary resonance complete tuning, as the electrostatic driving force amplitude increased. Both of the amplitudes of the first-order and second-order drive modes increased with the increase of the electrostatic force amplitude, and the growth relationship was nonlinear. The growth rate decreased gradually when the electrostatic force amplitude was large. The amplitude of the second-order drive mode reached saturation when the electrostatic force amplitude reached a certain threshold. In Figure 8b, the primary resonance was detuned, and the steady-state amplitudes of the first-order and second-order drive modes simultaneously exhibited a multiple stable solution phenomenon in the range of AB. Two of the three solutions were stable, and the initial conditions determined which solution was the real response. There was only one stable solution in other regions, which ws confirmed by numerical calculation. The amplitudes of the first-order and second-order drive modes were approximately linear with the electrostatic force amplitude when the electrostatic force amplitude was below the corresponding amplitude of point B. The amplitudes of the first-order and second-order drive modes jumped simultaneously when the electrostatic force amplitude reached point B. With jumping from the current amplitude to another higher stable amplitude, the amplitude of the second-order drive mode gradually reached saturation.

The first-order and second-order drive modes will produce vibrations regardless of the magnitude of the electrostatic driving force because of the stiffness of the first-order and second-order drive modes coupled with each other in structure. At the beginning, amplitudes grows linearly with the increase of electrostatic force amplitude. Then, the increase of driving amplitudes becomes nonlinear, and the phenomenon of the amplitude jump occurs when the primary resonance detuning. The amplitudes of the first-order and second-order drive modes increase slower with the increase in electrostatic force amplitude when it is higher than that of jumping region. Then, the amplitude of the second-order drive mode gradually reaches the saturation state. The response of the second-order drive mode is the output response of the driving direction, which determines the amplitudes of Coriolis force and sense mode responses. Therefore, the electrostatic force amplitude should be as large as possible when it is far away from the unstable region in order to make the output response of the gyroscope as large and stable as possible. At the same time, it should be noted that the driving voltage should not exceed the pull-in voltage of the comb electrodes.

## 4. Local Bifurcation Analysis

The local bifurcation response of the drive module was discussed for the further study of the influence of external disturbance on the dynamic performance of nonlinear gyroscope system. Here, the transformation of $a_{2}^{2}=Z$, *σ*_2_ = *μ* was done, and the recognition condition of the theory of singularity was used to simplify Equations (14) and (15). The local bifurcation equation of the drive direction can be obtained:(21)$$G(Z,\mu )=Z^{3}+g_{1}Z^{2}\mu +g_{2}Z\mu^{2}+g_{3}Z^{2}+g_{4}Z\mu +g_{5}\mu^{2}+g_{6}Z+g_{7}=0$$
where $g_{1}=\frac{h_{2}}{h_{1}},\;g_{2}=\frac{h_{3}}{h_{1}},\;g_{3}=\frac{h_{4}}{h_{1}},\;g_{4}=\frac{h_{5}}{h_{1}},\;g_{5}=\frac{h_{6}}{h_{1}},\;g_{6}=\frac{h_{7}}{h_{1}},\;g_{7}=\frac{h_{8}}{h_{1}}$. The specific parameters are given in Appendix A.

According to the theory of singularity, the transition sets in the unfolding parameter space is composed of Σ=B∪H∪D. Here, *B* is the bifurcation set, *H* is the hysteresis set, and *D* is the double limit point set. The specific expressions of the transition set are given in Appendix B.

Since the parameter space is a seven-dimensional space and the transition set cannot be visually expressed in space, the bifurcation behavior in the parameter space on the projection plane was mainly discussed. Here, *g*_6_, *g*_7_ were selected as the unfolding parameters. The transition sets and bifurcation diagrams on the *g*_6_–*g*_7_ plane were obtained, as shown in Figure 9.

On the projection plane *g*_6_–*g*_7_, the curves represented by the bifurcation set *B* and the hysteresis set *H* divide the plane into five sub-intervals. The bifurcation diagrams corresponding to two points on each sub-interval are topological equivalent, that is, persistent. Any two points that are not in the same sub-interval are topologically inequivalent. Taking a random point in each sub-interval, the retentive bifurcation diagram in each region was obtained, as shown in Figure 9 I to V. The system presents a more complicated bifurcation behavior with the change of the unfolding parameters *g*_6_, *g*_7_.

There is a unique parameter *Z* corresponding to the determined parameter *μ*, and the system is stable in this condition in the interval I and II. There are many values of *Z* corresponding to the determined parameter *μ*, and the phenomenon of multiple solutions occurs within intervals III, IV, and V. The system exhibits bifurcation behavior and periodic motion become instable in this moment. In interval III, parameter *Z* will change from the single solution region to the multi-solution region, when parameter *μ* changes from the value far from 0 to the vicinity of 0. In this case, the motion behavior of the system will change greatly, that is, the instability region is near 0. In intervals IV and V, parameter *Z* starts with only a unique solution, then reaches the first multi-solution region, becomes the only solution, passes through the second multi-solution region, and finally becomes the stable solution, with the *μ* becoming larger gradually. At present, there are two unstable regions in the system. In summary, the dynamic response of the system changes from the initial stable state to the unstable state, in which unstable regions exist, when the unfolding parameters *g*_6_, *g*_7_ caused by external disturbance change from interval I or II to interval III, IV, or V. The movement of the gyroscope changes essentially, which results in the system not working properly.

It can be seen from Appendix B that the parameters *h*_7_, *h*_8_ can be converted into the following form:(22)$$\begin{array}{ll} h_{7}= & 1048576{\hat{\beta }}_{1}^{4}\{{\hat{\alpha }}_{1}^{2}{\left({\hat{\beta }}_{2}-{\hat{\omega }}_{2}^{2}\right)}^{4}+{\hat{\omega }}_{1}^{2}{\left({\hat{\beta }}_{2}-{\hat{\omega }}_{2}^{2}\right)}^{3}[2{\hat{\alpha }}_{2}{\hat{\beta }}_{1}+{\hat{\omega }}_{1}^{2}({\hat{\beta }}_{2}-{\hat{\omega }}_{2}^{2})]\\ & -2{\hat{\alpha }}_{1}[{\hat{\alpha }}_{2}{\hat{\beta }}_{1}{\left({\hat{\beta }}_{2}-{\hat{\omega }}_{2}^{2}\right)}^{3}+{\hat{\omega }}_{1}^{2}{\left({\hat{\beta }}_{2}-{\hat{\omega }}_{2}^{2}\right)}^{3}-6{\hat{\beta }}_{2}^{2}{\hat{\omega }}_{1}^{2}{\hat{\omega }}_{2}^{4}]\}\\ h_{8}= & -1048576{\hat{\beta }}_{1}^{4}f^{2}[{\hat{\beta }}_{1}^{2}{\left({\hat{\beta }}_{2}-{\hat{\omega }}_{2}^{2}\right)}^{2}-6{\hat{\beta }}_{2}{\hat{\beta }}_{3}^{2}\omega_{0}^{2}{\hat{\omega }}_{2}^{2}] \end{array}$$

It can be seen from Equation (22) that almost every item in parameter *h*_7_ contains factor ${\hat{\beta }}_{1}^{4}({\hat{\beta }}_{2}-{\hat{\omega }}_{2}^{2})$. Therefore, the value of *h*_7_ is mainly affected by ${\hat{\beta }}_{1},{\hat{\beta }}_{2}$, and the stiffness coefficients *k*_2_ and *k*_3_ play major roles in the disturbance of parameter *g*_7_. The common factor in parameter *h*_8_ is ${\hat{\beta }}_{1}^{4}f^{2}$, so it can be seen that the stiffness coefficient *k*_2_ and the electrostatic driving force amplitude *F* play major roles in the disturbance of parameter *g*_8_. To keep the system as robust as possible to the environmental changes and avoid the occurrence of multiple solutions, the parameters *g*_6_, *g*_7_ should be taken in interval I or II. For this reason, the value of parameters *g*_6_, *g*_7_ could be increased as much as possible and kept at a positive value. That is to say, the stiffness coefficients *k*_2_ and *k*_3_ of the micro-beams should be increased to ensure the value of *g*_6_. To ensure the value of *g*_7_, the electrostatic driving force amplitude *F* and the damping coefficient *c*_2_ should be increased.

The unfolding parameters were used here to characterize the disturbance of the environment. Here, the environmental disturbance is a general concept which contains many factors, such as temperature, electrical, external force, and noise. Temperature noise, electrical noise, and phase noise [43,44] in noise mixing will affect the stability of the system from the system parameters or excitation signals. By adjusting the system parameters *k*_2_, *k*_3_, *c*_2_, *F*, the system could be placed in a relatively stable area as far as possible. From a qualitative point of view, it can suppress the influence of noise mixing.

## 5. Conclusions

In this paper, a 4-DOF micro gyroscope model including double drive-mode and double sense-mode was investigated. Particularly, the complex dynamic characteristics of the micro gyroscope were studied by considering the cubic stiffness nonlinearity, which may occur in both degree-of-freedom in the drive direction. The complex dynamic behaviors of the highly robust micro gyroscope system under nonlinear influence were obtained. The key conclusions are as follows:(1)The dynamic characteristics of the drive and sense modes are basically unaffected when the driving nonlinearity is relatively weak, which is similar to that of the linear design. However, the dynamic characteristics of the drive and sense modes will be obviously affected when the driving nonlinearity is strong. There are obvious amplitude jumps, multiple steady state solutions, and resonance frequency offsets in the amplitude-frequency response curves. Because the bandwidth regions of drive and sense modes match each other and have high robustness, the nonlinear hardening behavior in second-order drive mode and sense modes is no longer typical right bending but is instead right-down bending. Amplitude jumps and unstable solutions occur outside the response bandwidth of drive and sense modes and have only a slight effect on the sensitivity within the bandwidth.(2)When the internal resonance is close to the tuning state, part of the energy generated by the primary resonance of the first-order drive mode is transferred to the second-order drive mode. The response amplitude of the second-order drive mode increases gradually, while that of the first-order drive mode decreases. When the degree of internal resonance detuning increases, the energy transfer is relatively small, the response amplitude of the second-order drive mode decreases gradually, and the first-order drive mode increases slowly. The frequency drift caused by the change of the external environment will change the internal resonance parameter of the system. Even at a constant excitation frequency, the response amplitude will change or even change greatly. Therefore, external working conditions and processing errors should be fully considered, and corresponding error compensation should be made.(3)When the primary resonance complete tuning, the motions of 2-DOF drive mode are stable periodic motions with different electrostatic force amplitudes. When the main resonance is detuned, the response amplitudes of the drive modes will jump, and the multi-stable solution will appear with the increase of the electrostatic force amplitude and reach saturation when the electrostatic force amplitude increases to a certain value. In order to satisfy the stable output response of the gyroscope as much as possible, the electrostatic force amplitude should be as large as possible when it is far away from the unstable region. At the same time, it should be noted that the driving voltage must not exceed the pull-in voltage of the comb electrodes.(4)Through the study of the local bifurcation of the nonlinear gyroscope system, it is found that the stiffness coefficient of the micro-beams connected with the decoupling frame and the electrostatic driving force amplitude play major roles in the parameter perturbation. In order to keep the system as robust as possible to the environment and avoid the occurrence of multiple solutions phenomena, the stiffness coefficients of the micro-beams on both sides of the decoupling frame should be increased, and the damping coefficient between the driving mass and the decoupled frame and the electrostatic driving force amplitude should be appropriately increased.

## Figures and Tables

**Figure 1 micromachines-10-00578-f001:**
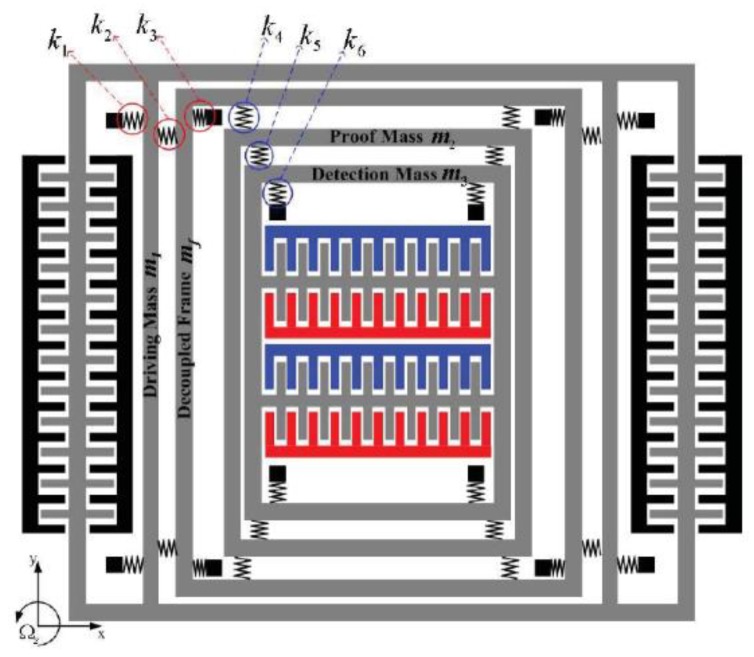
Structural schematic diagram of the micro gyroscope with two-degree-of-freedom (2-DOF) drive-mode and sense-mode.

**Figure 2 micromachines-10-00578-f002:**
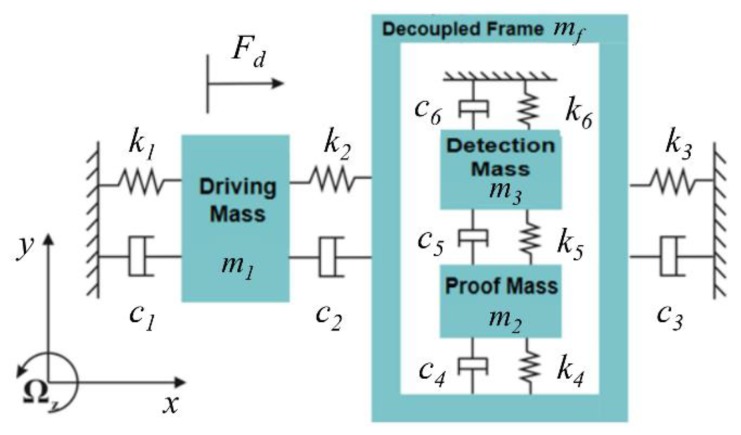
Lumped parameter model of the four-degree-of-freedom (4-DOF) micro gyroscope.

**Figure 3 micromachines-10-00578-f003:**
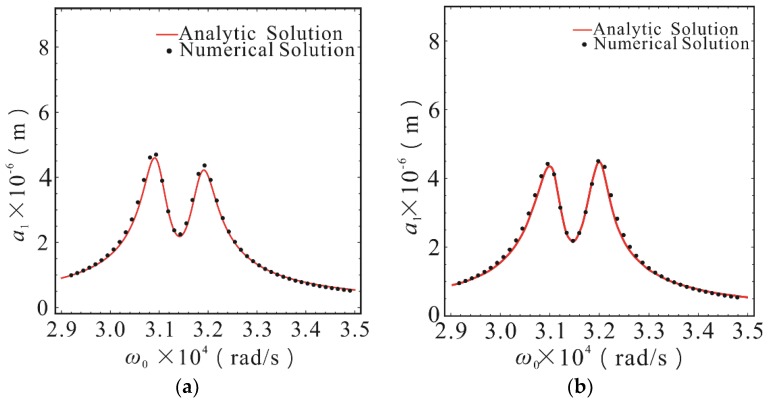
The amplitude-frequency response curves of first-order drive mode with different stiffness nonlinear coefficients in the case of *σ*_2_ = 5.91 × 10^7^. (**a**) The case of *K*_1_ = *K*_2_ = 0; (**b**) The case of *K*_1_ = *K*_2_ = 10^11^ N/m^3^; (**c**) The case of *K*_1_ = *K*_2_ = 10^11.5^ N/m^3^; (**d**) The case of *K*_1_ = *K*_2_ = 10^12^ N/m^3^. The point was calculated using the Runge-Kutta method. The line was calculated using the multiple scales method (MMS). The solid lines represent the stable solution. The dashed lines represent the unstable solution.

**Figure 4 micromachines-10-00578-f004:**
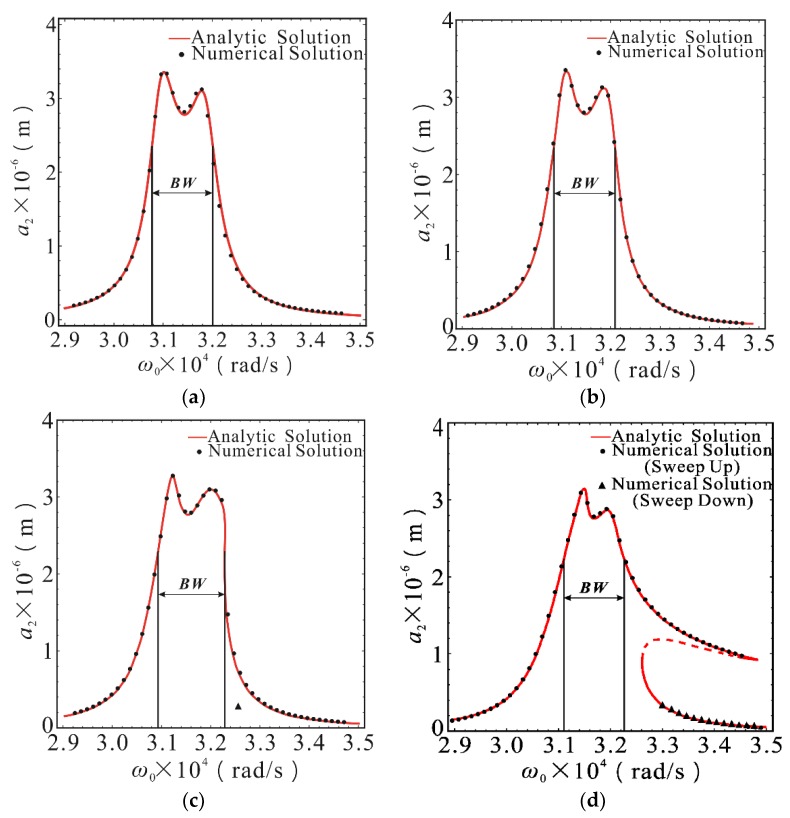
The amplitude-frequency response curves of second-order drive mode with different stiffness nonlinear coefficients in the case of *σ*_2_ = 5.91 × 10^7^. (**a**) The case of *K*_1_ = *K*_2_ = 0; (**b**) The case of *K*_1_ = *K*_2_ = 10^11^ N/m^3^; (**c**) The case of *K*_1_ = *K*_2_ = 10^11.5^ N/m^3^; (**d**) The case of *K*_1_ = *K*_2_ = 10^12^ N/m^3^. The point was calculated using the Runge-Kutta method. The line was calculated using the MMS. The solid lines represent the stable solution. The dashed lines represent the unstable solution. The *BW* area represents the 3dB bandwidth of the response.

**Figure 5 micromachines-10-00578-f005:**
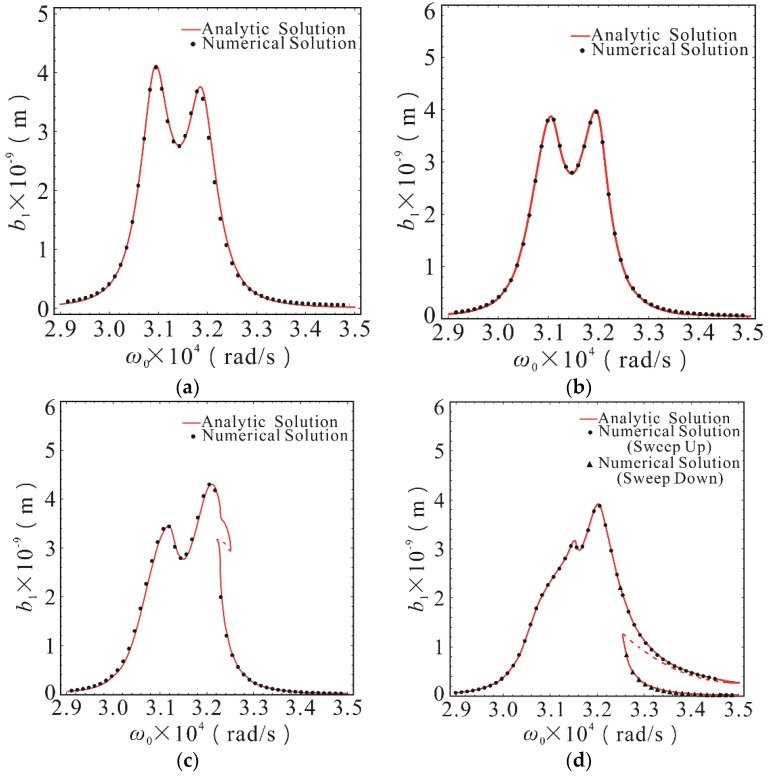
The amplitude-frequency response curves of first-order sense mode with different stiffness nonlinear coefficients in the case of *σ*_2_ = 5.91 × 10^7^. (**a**) The case of *K*_1_ = *K*_2_ = 0; (**b**) The case of *K*_1_ = *K*_2_ = 10^11^ N/m^3^; (**c**) The case of *K*_1_ = *K*_2_ = 10^11.5^ N/m^3^; (**d**) The case of *K*_1_ = *K*_2_ = 10^12^ N/m^3^. The point was calculated using the Runge-Kutta method. The line was calculated using the MMS. The solid lines represent the stable solution. The dashed lines represent the unstable solution.

**Figure 6 micromachines-10-00578-f006:**
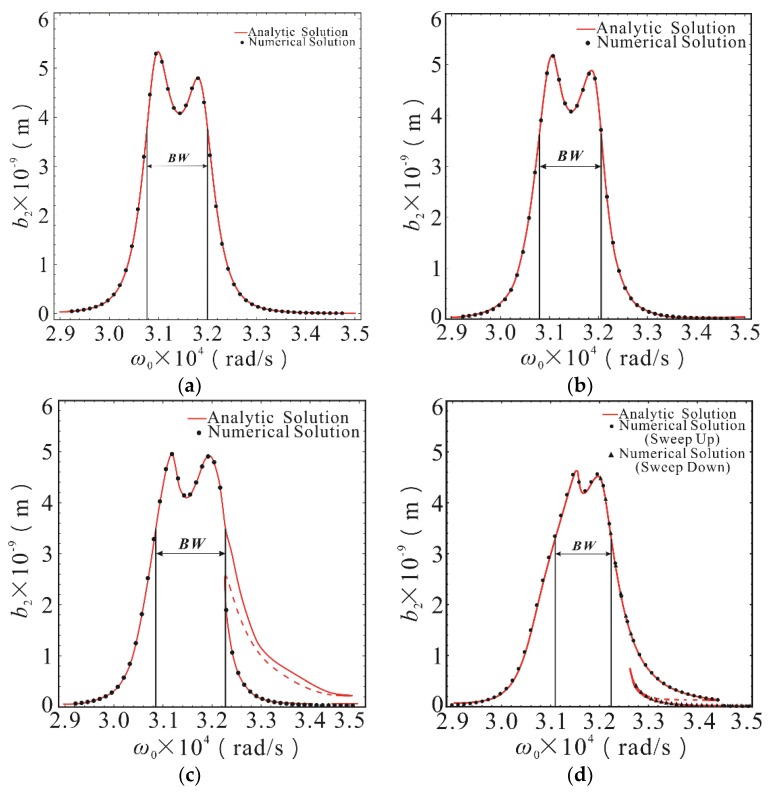
The amplitude-frequency response curves of second-order sense mode with different stiffness nonlinear coefficients in the case of *σ*_2_ = 5.91 × 10^7^. (**a**) The case of *K*_1_ = *K*_2_ = 0; (**b**) The case of *K*_1_ = *K*_2_ = 10^11^ N/m^3^; (**c**) The case of *K*_1_ = *K*_2_ = 10^11.5^ N/m^3^; (**d**) The case of *K*_1_ = *K*_2_ = 10^12^ N/m^3^. The point was calculated using the Runge-Kutta method. The line was calculated using the MMS. The solid lines represent the stable solution. The dashed lines represent the unstable solution. The *BW* area represents the 3dB bandwidth of the response.

**Figure 7 micromachines-10-00578-f007:**
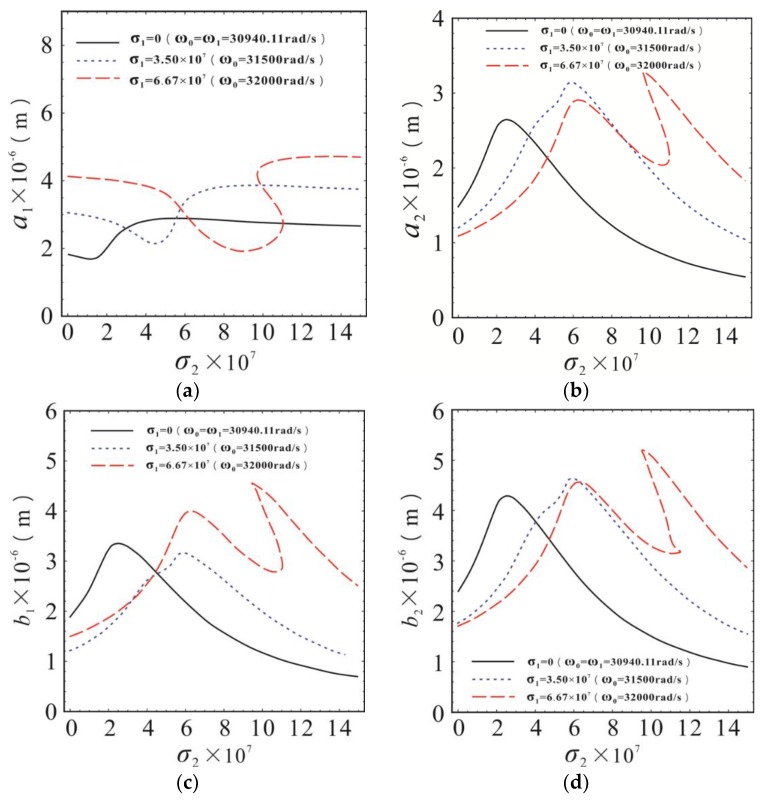
The variation curves of the response amplitude of each degree of freedom of the system with *σ*_2_ at different operating frequency *ω*_1_. (**a**) Relationship between response amplitude of the first-order drive mode and *σ*_2_; (**b**) Relationship between response amplitude of the second-order drive mode and *σ*_2_; (**c**) Relationship between response amplitude of the first-order sense mode and *σ*_2_; (**d**) Relationship between response amplitude of the second-order sense mode and *σ*_2_. The solid lines represent the stable solution. The dashed lines represent the unstable solution.

**Figure 8 micromachines-10-00578-f008:**
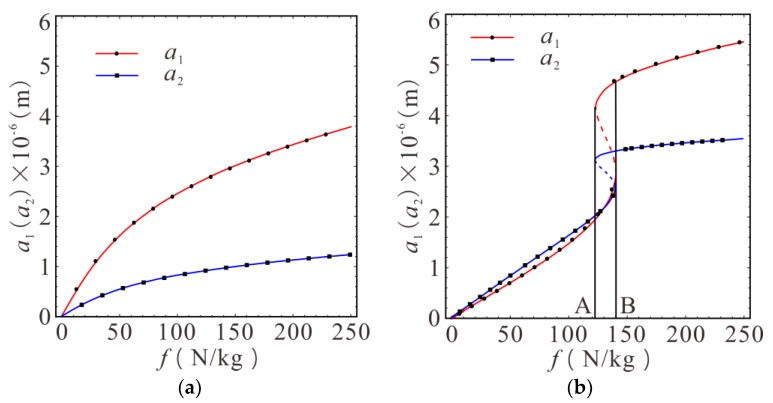
Relationship between the amplitude of the first-order and second-order drive modes and the electrostatic force amplitude. (**a**) The case of *σ*_1_ = 0; (**b**) The case of *σ*_1_ = 6.67 × 10^7^. The point was calculated using the numerical solution. The line was calculated using the MMS. The solid lines represent the stable solution. The dashed lines represent the unstable solution.

**Figure 9 micromachines-10-00578-f009:**
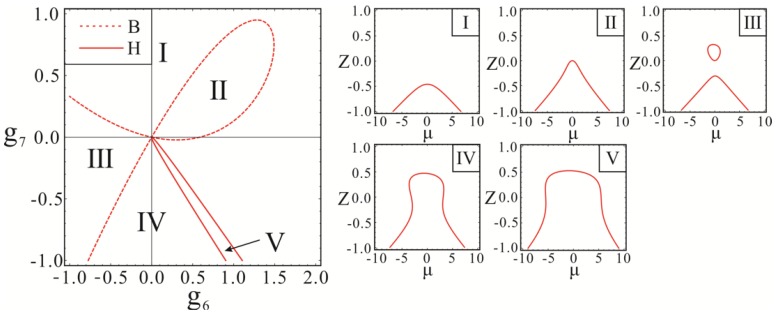
The transition sets and bifurcation diagrams on the *g*_6_–*g*_7_ plane.

**Table 1 micromachines-10-00578-t001:** The values of the physical parameters of the gyroscope.

Mass.	Value (kg)	Damping Coefficient	Value (Ns/m)	Stiffness Coefficient	Value (N/m)
*m* _1_	2.36 × 10^−7^	*c* _1_	10^−4^	*k* _1_	224.7
*m* _2_	2.74 × 10^−7^	*c* _2_	5 × 10^−6^	*k* _2_	8.2
*m_f_*	5.23 × 10^−8^	*c* _3_	2 × 10^−4^	*k* _3_	313.8
*m* _3_	1.35 × 10^−7^	*c* _4_	10^−4^	*k* _4_	260.9
-	-	*c* _5_	5 × 10^−6^	*k* _5_	9.5
-	-	*c* _6_	2 × 10^−4^	*k* _6_	123.7

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
