# Peer review of "Parametric Characteristics and Bifurcation Analysis of Multi-Degree-of-Freedom Micro Gyroscope with Drive Stiffness Nonlinearity"

_micromachines, 2019, doi:10.3390/mi10090578_

Round 1
Reviewer 1 Report
The paper studies the nonlinear dynamics of a 4-dof vibratory gyroscope system with the effects of both internal and external resonance parameters considered, and aims to establish guidelines for future designs of such systems. The topic is very interesting to the field and the analysis has been presented appropriately overall. However, I have a couple of questions regarding the nonlinear dynamic equations of the gyro system considered in the manuscript, which should be validated before acceptance.
First, the authors stated that the vacuum packaged device with low damping is considered and nonlinear damping is neglected, but the damping coefficient cited from Ref [13] is for gyro operated at atmospheric pressure. How does the values of the damping coefficient affect nonlinear system dynamics, qualitatively and quantitively?
Secondly, equations (3) did not include nonlinear interaction between the two masses related to (x1-x2), for example, nonlinear stiffness coefficient K_2 is not included in the first equation. How can this be justified? And how would it affect the results.
In addition, I have a few minor comments:
I feel some of the terminology used in the manuscript should be changed or better defined. For example, the use of “mode one/two” is not exactly accurate. The 2-dof coupled resonator has two resonant modes, a higher frequency mode and a lower frequency mode, which can be confused with the mode one and mode two the authors used in the paper. I think the what the author are referring to is the displacement of the mass one and mass two under the two resonant modes. Another example is the use of “sensitivity” (to drive force?) in the discussion of the drive mode dynamics, which can also be confusing in the context of gyroscopes.
For figures 3 to 6, since the authors are quantitatively comparing the results under different nonlinearity conditions, maybe overlaying the (a)(b)(c)(d) subplots can better visualize the differences.
Reviewer 2 Report
Report on paper "Parametric characteristics and bifurcation analysis of multi-degree-of-freedom micro gyroscope with drive stiffness nonlinearity" submitted by Han et al., for publication in Micromachines (micromachines-567202).
The authors developed an analytical model to investigate the dynamic behavior of a 4-DOF micro gyroscope with double drive-mode and double sense-mode, while taking into account the cubic stiffness nonlinearity in both degree-of-freedom in the drive direction. A parametric study is carried out to investigate the influence of nonlinearity on the sensor specifications as well as its robustness. Although the topic of this work is interesting and the authors showed some promising results, the paper must benefit from a potential improvement by addressing the following specific comments:
The abstract should be concise and clear with a focus on the originality and highlights of the paper. The literature survey lacks of references in the field of nonlinear MEMS gyroscopes, which is a topic deeply investigated even in the recent past (for instance *Sensors and Actuators A: Physical Volume 177, 79-86, 2012, **International Journal of Non-Linear Mechanics Volume 46, Issue 10, 1347-1355, 2011, *** Scientific Reports volume 5, 9036, 2015). The authors should highlight clearly the originality of their work with respect to the literature, in particular with respect to reference 13. The assumptions for using MMS should be stated and the scaling of Equation (6) should be justified. In line 196, "primary resonance" instead of "primary parametric resonance". The bandwidth enhancement occurs essentially in the linear dynamic range. It is not clear what role the nonlinearity has at this point? It would be interesting if the authors specify the level of amplitude at which the gyroscope can be driven in a linear fashion compared to the critical amplitude (amplitude beyond which bistability occurs). When the gyroscope operates in the nonlinear regime, noise-mixing phenomenon will drastically deteriorate the sensor resolution [IEEE transactions on ultrasonics, ferroelectrics, and frequency control, vol. 52, no. 12, december 2005]. This point should be discussed with respect to relevant references from the literature dealing with performance enhancement of MEMS resonant sensors (for instance *J. Micromech. Microeng. 18(6):065014, 2008, **From MEMS to NEMS: Closed-loop actuation of resonant beams beyond the critical Duffing amplitude, 2008 IEEE Sensors, Lecce, 2008, pp. 510-513, ***Appl. Phys. Lett. 88, 253101, 2006, **** Nanotechnology 20(27):275501, 2009). In the conclusion, the authors said that the damping should be increased to keep the system as robust as possible to the environment. However, this will have an impact on the gyroscope sensitivity, and possibly there is a compromise between robustness and high sensitivity. The authors should discuss this point. The quality of all figures must be enhanced.Author Response
Please see the attachment.

Round 2
Reviewer 1 Report
I appreciate the response and revisions from the authors. This seems to be a better manuscript now.
The definition and usage of the term "modes" is still unclear. Simply changing "mode one/two" to "low frequency mode/high frequency mode" does not make much difference. For example, isn't the author plotting both "low frequency mode" response at 30904.11rad/s and "high frequency mode" response at 31880.86rad/s in either figure 3 or figure 4, but just for different DOF (unlike what the figure captions are stating)? The descriptions in the text as well as figure captions need to be modified to be clear.
Reviewer 2 Report
The authors did not address point 7 concerning noise mixing, which is a key problem for such sensors. I invite the authors to discuss the issue in a short paragraph while considering relevant references (see previous review) that tackled noise mixing and proposed strategies to avoid it. In the introduction "Sharma et al.", Kacem et al.", "Nitzan et al." instead of "Sharma, M et al.", Kacem, N et al.", "Nitzan, SH et al.".
